# ACCEPT MORE, REJECT LESS: REDUCING UP TO 19% UNNECESSARY DESK-REJECTIONS OVER 11 YEARS OF ICLR DATA

## ABSTRACT

The explosive growth of AI research has driven paper submissions at flagship AI conferences to unprecedented levels, necessitating many venues in 2025 (e.g., CVPR, ICCV, KDD, AAAI, IJCAI, WSDM) to enforce strict per-author submission limits and to desk-reject any excess papers by simple ID order. While this policy helps reduce reviewer workload, it may unintentionally discard valuable papers and penalize authors' efforts. In this paper, we ask an essential research question on whether it is possible to follow submission limits while minimizing needless rejections. We first formalize the current desk-rejection policies as an optimization problem, and then develop a practical algorithm based on linear programming relaxation and a rounding scheme. Under extensive evaluation on 11 years of real-world ICLR (International Conference on Learning Representations) data, our method preserves up to 19.23% more papers without violating any author limits. Moreover, our algorithm is highly efficient in practice, with all results on ICLR data computed within at most 53.64 seconds. Our work provides a simple and practical desk-rejection strategy that significantly reduces unnecessary rejections, demonstrating strong potential to improve current CS conference submission policies.

## 1 INTRODUCTION

The era of artificial intelligence (AI) is rapidly unfolding, with numerous breakthroughs across a wide range of real-world applications. Notable examples include visual generation (Song et al., 2021; Ho et al., 2022; Blattmann et al., 2023), language reasoning (Schulman et al., 2022; Achiam et al., 2023; Anthropic, 2024), and automated drug discovery (Senior et al., 2020; Jumper et al., 2021; Abramson et al., 2024). A key driving force behind the progress of AI is the set of major conferences held annually,

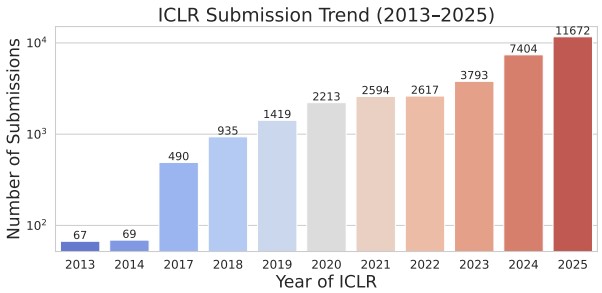

Figure 1: **ICLR Submission Trend (2013–2025).** Total number of paper submissions to ICLR each year from 2013 to 2025, plotted on a log scale. Data for 2015 and 2016 are omitted due to missing records in OpenReview.

which present the latest research developments and serve as a platform for researchers to exchange ideas. The success of AI is deeply rooted in the research papers published at these venues. Influential works such as ResNet (He et al., 2016), Transformers (Vaswani et al., 2017), BERT (Devlin et al., 2019), diffusion models (Ho et al., 2020), and CLIP (Radford et al., 2021) have laid the foundation for modern AI models, received widespread citations, and fundamentally transformed the field.

The growing interest in AI research has led to a rapid increase in the number of AI-related publications in recent years (e.g., the sharp rise in ICLR submissions shown in Figure 1), as documented by recent studies (Stanford, 2024; Allen-Zhu & Xu, 2025; Cao et al., 2025). A direct consequence of this growth is the substantial rise in paper submissions to major AI conferences, which has placed a

considerable burden on program committees responsible for selecting high-quality work. To address this challenge and preserve the quality of accepted papers, many top-tier conferences have introduced submission limits per author. In recent years, several leading conferences, including CVPR, ICCV, AAAI, WSDM, IJCAI, KDD, and ICDE adopted author-level submission upper bounds ranging from $b = 7$ to $b = 25$ (see Table 1 for a summary). These policies typically reject all submissions that include at least one author exceeding the limit, often retaining only those papers with the smallest submission IDs (e.g., see CVPR 2025 website[1]).

Despite the simplicity and effectiveness of current submission-limit-based desk-rejection policies, desk-rejecting a large number of papers before the review stage may have substantial negative consequences for affected authors. In today's highly competitive AI job market, publication records play a critical role in shaping researchers' careers. Motivated by this, we investigate the following core question: **Can we design a submission-limit-based desk-rejection policy that minimizes the number of desk-rejected papers while satisfying author-level submission constraints?**

This research question is highly important, as it goes beyond existing straightforward policies by seeking a principled approach that better preserves authors' efforts while still maintaining the review workload within acceptable bounds. Our goal is to achieve a balance between keeping more papers for authors and complying with the submission limits from conference organizers.

Table 1: In this table, we summarize the submission limits of recent top conferences. We denote the absence of a limit with "N/A". The full list of conference submission limit policy URLs is provided in Appendix D.

| Conference Name | Year | Submission Limit |
|---|---|---|
| CVPR | 2025 | 25 |
| CVPR | 2024 | N/A |
| ICCV | 2025 | 25 |
| ICCV | 2023 | N/A |
| AAAI | 2023-2025 | 10 |
| AAAI | 2022 | N/A |
| WSDM | 2021-2025 | 10 |
| WSDM | 2020 | N/A |
| IJCAI | 2021-2025 | 8 |
| IJCAI | 2020 | 6 |
| IJCAI | 2018-2019 | 10 |
| IJCAI | 2017 | N/A |
| KDD | 2024-2025 | 7 |
| KDD | 2023 | N/A |
| ICDE | 2025 | 10 |
| ICDE | 2024 | N/A |

To re-examine existing submission-limit-based desk-rejection policies and explore potential improvements, we introduce a formal mathematical formulation of the submission-limit problem in AI conferences. We first model current desk-rejection rules used by major AI venues, and then propose a novel re-formulation of the problem as a discrete optimization task. From a technical perspective, we establish the computational hardness of the problem and design an efficient algorithm based on linear programming relaxation and randomized rounding. We further validate our approach using real-world data from 11 years of ICLR submissions. Our contributions are summarized as follows:

- We develop a rigorous mathematical framework that formalizes existing submission-limit-based desk-rejection policies adopted by AI conferences.

- We propose a novel optimization-based desk-rejection mechanism that satisfies submission limits while maximizing author welfare by preserving more submitted papers.

- We conduct extensive empirical evaluations on 11 years of ICLR submission data and show that our method consistently outperforms existing policies in all cases, reducing the number of desk-rejections by up to $19.23\%$. In practice, our algorithm is highly efficient, with all results on ICLR data obtained within no more than 53.64 seconds.

**Roadmap.** In Section 2, we present the relevant works. In Section 3, we show the formal definition of the submission-limit policies in AI conferences. In Section 4, we outline our proposed method for reducing unnecessary desk rejections. In Section 5, we present the empirical results. In Section 6, we conclude our paper.

## 2 RELATED WORKS

**Desk-Rejection Mechanisms.** Various desk-rejection mechanisms have been proposed to reduce reviewer workload in peer review (Ansell & Samuels, 2021), including violations of anonymity (Jef-

---

[1]https://cvpr.thecvf.com/Conferences/2025/CVPRChanges

ferson et al., 2002; Tennant, 2018), dual submissions (Stone, 2003; Leopold, 2013), and plagiarism (King & ChatGPT, 2023; Elali & Rachid, 2023). With the surge in AI submissions, new policies have emerged (Leyton-Brown et al., 2024), such as rapid abstract-based rejection at IJCAI 2020 (Committees, 2020a) and NeurIPS 2020 (Committees, 2020b). This paper studies a more recent policy: desk rejection based on exceeding per-author submission limits (see Table 4 in Appendix D for a list of conferences with this rule). To our knowledge, prior work on this mechanism is scarce, and we provide one of the first formal formulations and optimization-based analyses.

**The Competitive Race in AI Publication.** The rapid increase in paper submissions to AI conferences in recent years (Stanford, 2024) has raised growing concerns about the intensifying competitiveness of the field. As noted by Bengio (Bengio, 2020): "It is more competitive, everything is happening fast and putting a lot of pressure on everyone. The field has grown exponentially in size. Students are more protective of their ideas and in a hurry to put them out, by fear that someone else would be working on the same thing elsewhere, and in general, a PhD ends up with at least 50% more papers than what I gather it was 20 or 30 years ago." In this environment, publication records have become increasingly important for job applications in academia and industry (Ahmed, 2022; Besiroglu et al., 2024; Allen-Zhu & Xu, 2025), with a higher number of accepted papers now considered standard. As a result, it is essential to establish better desk-rejection policies (Teixeira da Silva et al., 2018) to ensure author welfare in the AI research community.

Due to space limits, we defer a more comprehensive review of related works to Appendix A.

## 3 SUBMISSION-LIMIT-BASED DESK REJECTION

We begin by introducing some useful notations that will be used later. Next, we formulate the submission-limit-based desk rejection in Section 3.1, and then show existing solutions in Section 3.2.

**Notations.** We use $[n]$ to denote the set of positive integers $\{1, 2, \cdots, n\}$. Let $n \in \mathbb{N}_+$ denote the number of authors and $m \in \mathbb{N}_+$ denote the number of papers. Let $b \in \mathbb{R}$ be the maximum number of submissions for each author. Let $A \in \{0, 1\}^{n \times m}$ denote the authorship matrix, such that $A_{i,j} = 1$ denotes author $i$ is in the author list of paper $j$, and $A_{i,j} = 0$ otherwise. We use $\mathbf{1}_n$ to denote an $n$-dimensional column vector whose all entries are one.

### 3.1 PROBLEM FORMULATION

In recent AI conferences, due to the surge in the number of submissions, it is hard to assign knowledgeable reviewers to these papers. To improve the review quality and reduce the conference organization load, AI conferences impose submission limits. Specifically, the submission limit problem can be formulated as:

**Definition 3.1** (Submission limit problem). *For a specific number of authors $n$, number of papers $m$, and authorship matrix $A \in \{0, 1\}^{n \times m}$, we seek to find a desk-rejection vector $x \in \{0, 1\}^m$ to satisfy the submission limit $b$, such that*

$$\sum_{j=1}^m A_{i,j} x_j \le b, \quad \forall i \in [n].$$

This problem finds a feasible solution to satisfy the submission limit, while it does not optimize the number of papers that are desk-rejected.

We also define the concept of desk-rejection and desk-acceptance, which is useful for describing the algorithms in this paper.

**Definition 3.2** (Desk-rejection and Desk-acceptance). *Consider a solution $x \in \mathbb{R}^m$ as Definition 3.1. For an arbitrary $j \in [m]$, we say that paper $j$ is **desk-rejected** if $x_j = 0$, and we say that paper $j$ is **desk-accepted** if $x_j = 1$.*

**Remark 3.3.** *The notions of desk-rejection and desk-acceptance differ from traditional paper rejection and acceptance. They only enforce compliance with author-level submission limits and determine which papers are forwarded to the peer review process.*

---

**Algorithm 1** Current desk-reject algorithm that rejects all paper at once

---

1: **procedure** ALLREJECT($A \in \{0,1\}^{n \times m}, n, m, b \in \mathbb{N}_+$)
2:     $R \leftarrow \emptyset$                                     $\triangleright$ The set of papers to be desk-rejected
3:     **for** $i \in [n]$ **do**
4:         $P_i \leftarrow \{j : j \in [m], A_{i,j} = 1\}$    $\triangleright$ All papers that include $i$ as an author. This step takes $O(k_1)$ time.
5:         **if** $|P_i| > b$ **then**
6:             Find a set $R_i \subseteq P_i$ such that $|R_i| = |P_i| - b$
7:             $R \leftarrow R \cup R_i$
8:         **end if**
9:     **end for**
10:    $x \leftarrow \mathbf{1}_m$                                       $\triangleright$ The desk-rejection result
11:    **for** $j \in R$ **do**
12:       $x_j \leftarrow 0$
13:    **end for**
14:    **return** $x$                                      $\triangleright x \in \{0,1\}^m$
15: **end procedure**

---

## 3.2 EXISTING DESK-REJECTION ALGORITHMS

Currently, AI conferences adopt a straightforward desk-rejection mechanism that checks each author and rejects all papers exceeding the submission limit. We formalize this policy in Algorithm 1. To select the subset of papers to be rejected (line 6), conferences often prioritize papers with higher submission IDs, as reflected in many official conference guidelines (e.g., the CVPR 2025 guideline).

For convenience in time complexity analysis, we define the following parameters:

**Definition 3.4** (Maximum paper count and maximum author count). *Let $k_1 := \max_{i \in [n]} |\{j \in [m] : A_{i,j} = 1\}|$ denote the maximum number of papers authored by any single author, and let $k_2 := \max_{j \in [m]} |\{i \in [n] : A_{i,j} = 1\}|$ denote the maximum number of authors on any single paper.*

Algorithm 1 is guaranteed to solve the submission limit problem in Definition 3.1, as stated below:

**Proposition 3.5** (Correctness of ALLREJECT, informal version of Proposition B.1). *In $O(nk_1)$ time, Algorithm 1 produces a feasible solution for $x$ as defined in Definition 3.1.*

The current implementation of the desk-rejection policy inevitably rejects more papers than necessary, beyond each author's excess submissions (i.e., $|P_i| - b$), since it rejects all excessive papers at once. A more careful implementation would sequentially check each paper based on its submission count while dynamically maintaining a per-author paper counter.

This approach leads to two equivalent algorithms: one that processes papers in forward order (Algorithm 2) and one in reverse order (Algorithm 5). Both serve as stronger baseline desk-rejection policies and are provably correct in producing feasible solutions. We present the forward version here (Algorithm 2) and formally establish its correctness and time complexity in Proposition 3.6. The reverse version, due to its equivalence to the forward version, is deferred to Appendix C.

**Proposition 3.6** (Correctness of FORWARDREJECT, informal version of Proposition B.2). *Algorithm 2 takes $O(mk_2)$ time and outputs the feasible solution for $x$ as defined in Definition 3.1.*

**Remark 3.7.** *The time complexity of all the conventional desk-rejection algorithms (Algorithms 1, 2, and 5) can also be written as $O(\mathrm{nnz}(A))$, replacing both the $O(nk_1)$ and $O(mk_2)$ terms.*

## 4 MINIMIZING UNNECESSARY DESK REJECTIONS

In Section 4.1, we formulate the submission-limit-based desk-rejection system as maximum desk-acceptance submission limit problem. In Section 4.2, we present an efficient two-stage solver: we solve a linear-program relaxation of the integer program and subsequently apply a specific rounding algorithm to recover a provably feasible integer solution.

---

**Algorithm 2** Current desk-reject algorithm that "accepts" safe papers sequentially

---

1: **procedure** FORWARDREJECT($A \in \{0,1\}^{n \times m}, n, m, b \in \mathbb{N}_+$)
2: $\quad P_i \leftarrow \emptyset, \forall i \in [n]$ $\quad\quad\quad\quad\quad\quad$ ▷ The set of desk-accept paper for each author $i$
3: $\quad$ **for** $j = 1 \ldots m$ **do**
4: $\quad\quad S \leftarrow \{i : i \in [n], A_{i,j} = 1\}$ $\quad$ ▷ All the authors for paper $j$. This takes $O(k_2)$ time.
5: $\quad\quad$ **if** $|P_i| < b, \forall i \in S$ **then** ▷ We desk-accept a paper if all its authors are not reaching the limit. Check this takes $O(k_2)$ time.
6: $\quad\quad\quad$ **for** $i \in S$ **do**
7: $\quad\quad\quad\quad P_i \leftarrow P_i \cup \{j\}$
8: $\quad\quad\quad$ **end for**
9: $\quad\quad$ **end if**
10: $\quad$ **end for**
11: $\quad P \leftarrow \bigcup_{i=1}^n P_i$ $\quad\quad\quad\quad\quad\quad\quad\quad\quad$ ▷ All the accepted papers
12: $\quad x \leftarrow \mathbf{0}_m$ $\quad\quad\quad\quad\quad\quad\quad\quad\quad\quad$ ▷ The desk-rejection result
13: $\quad$ **for** $j \in P$ **do**
14: $\quad\quad x_j \leftarrow 1$
15: $\quad$ **end for**
16: $\quad$ **return** $x$ $\quad\quad\quad\quad\quad\quad\quad\quad\quad\quad\quad$ ▷ $x \in \{0,1\}^m$
17: **end procedure**

---

### 4.1 THE INITIAL INTEGER PROGRAMMING PROBLEM

The current submission-limit-based desk-rejection problem in Definition 3.1 produces a feasible solution that satisfies the submission limit but does not aim to reduce the number of desk-rejections and respect authors' efforts. Motivated by the utilitarian social welfare (Sen, 1979; Myerson, 1981; Boutilier et al., 2012; Aziz et al., 2024), we present an important reformulation of the submission-limit-based desk-rejection from an optimization perspective.

**Definition 4.1** (Maximum desk-acceptance submission limit problem). *Let $n$ be the number of authors, and $m$ be the number of papers. Let $A_{i,j} = 1$ if author $i \in [n]$ is an author of paper $j \in [m]$, and $A_{i,j} = 0$ otherwise. Let $b$ denote the maximum number of papers each author is allowed to submit. Let $x \in \mathbb{R}^m$ denote the desk-rejection vector, where each $x_j$ indicates whether paper $j$ is desk-rejected. We define the following optimization problem:*

$$\max_{x \in \{0,1\}^m} \mathbf{1}_m^\top x$$
$$\text{s.t. } Ax \leq b \cdot \mathbf{1}_n$$

In the definition above, the objective $\mathbf{1}_m^\top x$ maximizes the number of desk-accepted papers, while the constraint $Ax \leq b \cdot \mathbf{1}_n$ ensures that no author exceeds the submission limit. This formulation balances author welfare (i.e., desk-rejecting less papers) with conference submission constraints.

**Remark 4.2.** *The maximum desk-acceptance formulation of submission-limit-based desk rejection (Definition 4.1) clearly improves author welfare compared with the conventional formulation (Definition 3.1). The new formulation explicitly seeks an optimal solution that desk-rejects as few papers as possible, whereas the previous one only guarantees a feasible solution satisfying the submission limits without optimizing the number of desk rejections.*

### 4.2 LINEAR PROGRAM RELAXATION AND ROUNDING STRATEGY

The maximum desk-acceptance submission limit problem is a standard integer programming problem, inherently related to the multi-dimensional knapsack problem (Kellerer et al., 2004), and cannot be solved efficiently in general. To address this, we relax the domain of $x$ to $[0,1]^m$, allowing fractional solutions between desk-rejection and desk-acceptance.

**Definition 4.3** (Linear program relaxation of Definition 4.1). *Let $n$ be the number of authors and $m$ the number of papers. Let $A_{i,j} = 1$ if author $i \in [n]$ is an author of paper $j \in [m]$, and 0 otherwise. Let $b$ denote the maximum number of papers that each author may submit. Let $x \in \mathbb{R}^m$ denote the desk-rejection vector indicating whether each paper is accepted. We define the following*

*optimization problem:*

$$\max_{x \in [0,1]^m} \mathbf{1}_m^\top x$$

$$\text{s.t. } Ax \le b \cdot \mathbf{1}_n$$

**Remark 4.4.** *The time complexity to solve the relaxed maximum desk-acceptance problem in Definition 4.3 aligns with modern linear programming solvers. For instance, using the stochastic central path method (Cohen et al., 2021b; Jiang et al., 2021; Qin et al., 2023), it achieves $O^*(m^{2.37} \log(m/\delta))$ runtime[2], where $\delta$ represents the relative accuracy corresponding to a $(1+\delta)$-approximation guarantee.*

**Remark 4.5.** *In practice, major AI conferences commonly process submissions at the scale of $m \sim 10^4$ (Stanford, 2024). Given this fact, our algorithm guarantees efficient computation, enabling desk-rejection maximized desk rejection within feasible time resources, even for large-scale conferences.*

Despite the efficiency of solving the relaxed linear program, it may not yield the optimal solution of the original integer program. Cases where the relaxed linear program directly produces an integer solution are rare and occur only when the constraint matrix $A$ satisfies the total unimodularity (Schrijver, 1998; Beck & Guttmann-Beck, 2025). Therefore, we propose a rounding algorithm to handle the fractional solutions produced by the linear program in Definition 4.3, with guaranteed correctness. The proposed rounding algorithm is detailed in Algorithm 3.

---

**Algorithm 3** The rounding algorithm to convert the fractional solutions to integers

---

1: **procedure** MAXROUNDING($x \in [0,1]^m$, $A \in \{0,1\}^{n \times m}$, $b \in \mathbb{N}_+$)
2:      Let $S \leftarrow \{j \in [m] \mid x_j \in (0,1)\}$              $\triangleright$ The set $S$ has $O(m)$ elements
3:      **for** $i \in [n]$ **do**
4:          $T_i \leftarrow \{j \in [m] \mid A_{i,j} = 1\}$          $\triangleright$ $T_i$ is all the papers written by author $i$
5:      **end for**              $\triangleright$ Building all $m$ sets needs $O(nk_1)$ time
6:      Let $\widetilde{x} \leftarrow x$
7:      **while** $S \ne \emptyset$ **do**          $\triangleright$ $S$ denote the set of papers get fractional solution
8:          Let $l \leftarrow \arg\max_{j \in S} \widetilde{x}_j$
9:          $\widetilde{x}_l \leftarrow 1$          $\triangleright$ We will keep the paper $l$
10:          $S \leftarrow S \setminus \{l\}$
11:          Let $Q \subseteq [n]$ denote the set of authors of paper $l \in [m]$     $\triangleright$ Set $Q$ has $O(k_2)$ elements
12:          **for** $i \in Q$ **do**
13:              **if** $\sum_{j \in T_i} \widetilde{x}_j > b$ **then**
14:                  Find the set $S_i \subseteq (S \cap T_i)$ such that $\sum_{j \in S_i} \widetilde{x}_j \ge (1 - x_l)$   $\triangleright$ This takes $O(k_1)$ time.
15:                  For all $j \in S_i$, $\widetilde{x}_j \leftarrow 0$          $\triangleright$ We will desk-reject all of such papers
16:                  $S \leftarrow S \setminus S_i$
17:              **end if**
18:          **end for**              $\triangleright$ The for loop takes $O(k_1 k_2)$ time.
19:      **end while**
20:      **return** $\widetilde{x}$              $\triangleright$ $\widetilde{x} \in \{0,1\}^m$
21: **end procedure**

---

This rounding algorithm guarantees correctness in converting fractional solutions from the linear program into fully binary $\{0,1\}$ solutions. This ensures that the results can be directly used to determine paper desk-acceptance and desk-rejection decisions, without ambiguity or risk of violating the submission limit. The formal statement of correctness is presented below:

**Theorem 4.6** (Correctness of MAXROUNDING, informal version of Theorem B.3). *Algorithm 3 takes $O(nk_1 + mk_1k_2)$ time and outputs the feasible solution for $x$ as defined in Definition 3.1.*

Now, we combine the relaxation and rounding algorithms to obtain a unified desk-rejection framework that considers both author welfare and submission limits. This algorithm is presented in Algorithm 4. The overall time complexity of our algorithm is the sum of the time required to solve the linear program and the time complexity of the max-rounding procedure.

---

[2]The $O^*$-notation omits the $m^{o(1)}$ factors.

---

**Algorithm 4** Our proposed submission-limit-based desk-rejection algorithm

---

1: **procedure** OPTREJECT($A \in \{0,1\}^{n \times m}, n, m, b \in \mathbb{N}_+$)
2:      Randomly initialize $x_0$
3:      $x \leftarrow$ LPSOLVER($x_0, A, b$)      $\triangleright$ Solve the linear program in Definition 4.3 with any solver
4:      $\widetilde{x} \leftarrow$ MAXROUNDING($x, A, b$)      $\triangleright$ Call Algorithm 3
5:      **return** $\widetilde{x}$
6: **end procedure**

---

Table 2: **Statistics of the ICLR datasets.** MSPA (Max Submissions Per Author) denotes the maximum number of papers submitted by a single author, indicating the row sparsity of the authorship matrix $A$. ASPA (Average Submissions Per Author) denotes the average number of papers submitted per author. ICLR 2015 and ICLR 2016 are excluded due to missing data in the OpenReview system, which prevents us from crawling them.

| Year | # Authors ($n$) | # Papers ($m$) | nnz($A$) | MSPA | ASPA |
|------|------|------|------|------|------|
| 2013 | 161 | 67 | 190 | 7 | 2.40 |
| 2014 | 187 | 69 | 217 | 7 | 2.71 |
| 2017 | 1474 | 490 | 1825 | 8 | 3.01 |
| 2018 | 2820 | 935 | 3512 | 12 | 3.02 |
| 2019 | 4388 | 1419 | 5619 | 23 | 3.09 |
| 2020 | 6963 | 2213 | 9117 | 26 | 3.15 |
| 2021 | 7964 | 2594 | 10854 | 30 | 3.07 |
| 2022 | 8507 | 2617 | 11572 | 23 | 3.25 |
| 2023 | 12451 | 3793 | 17375 | 24 | 3.28 |
| 2024 | 23382 | 7404 | 35912 | 35 | 3.16 |
| 2025 | 38495 | 11672 | 61992 | 42 | 3.30 |

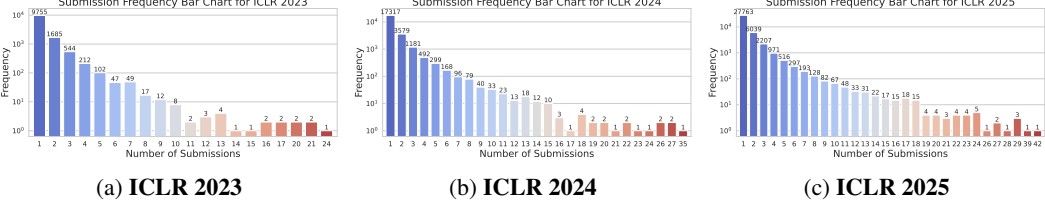

(a) **ICLR 2023**        (b) **ICLR 2024**        (c) **ICLR 2025**

Figure 2: **Submission Frequency Bar Charts for ICLR 2023–2025.**

## 5 EXPERIMENTS

In Section 5.1, we introduce the settings of our experiments. In Section 5.2, we present the extensive comparison results.

### 5.1 EXPERIMENTAL SETTINGS

**Datasets.** Desk-rejection data from most conferences (e.g., CVPR, KDD) is not public and can be accessed only by the conference chairs. ICLR is the only venue with public submission records, so we evaluate our method on ICLR data. We obtain the data through the OpenReview API[3] and run simulation experiments.

Specifically, we exaustively collect all ICLR submissions from 2013 to 2025. ICLR 2015 is excluded because no records are on OpenReview, and ICLR 2016 is excluded because its OpenReview page lists only workshop papers. For ICLR 2013–2014, we query OpenReview API v1 with the invitation link "`ICLR.cc/{year}/conference/-/submission`". For ICLR 2017–2023, we use OpenReview API v1 with the invitation link "`ICLR.cc/{year}/Conference/-/Blind`

---

[3]`https://docs.openreview.net/`

Table 3: **Number of Desk-Rejected Papers Across Desk-Rejection Methods.** Number of rejected papers for conventional desk-rejection methods and our proposed method at submission limits $b = 4$ to $b = 25$. Lower values indicate fewer papers desk-rejected. **N/A** denotes that no author has submitted more than $b$ papers, so all desk-rejection algorithms reject no papers. **Relative Improvement (%)** indicates the percentage improvement of our method over the strongest baseline.

| Dataset | Method | $b=4$ | $b=7$ | $b=10$ | $b=13$ | $b=16$ | $b=19$ | $b=22$ | $b=25$ |
|---|---|---|---|---|---|---|---|---|---|
| ICLR 2018 | ALLREJECT | 56 | 18 | 5 | 0 | 0 | 0 | 0 | 0 |
| ($n = 935$) | FORWARDREJECT | 53 | 18 | 5 | 0 | 0 | 0 | 0 | 0 |
| | **Ours** | **51** | **17** | **5** | **0** | **0** | **0** | **0** | **0** |
| | Relative Improvement (%) | 3.77% | 5.56% | 0.00% | N/A | N/A | N/A | N/A | N/A |
| ICLR 2019 | ALLREJECT | 127 | 43 | 18 | 11 | 7 | 4 | 1 | 0 |
| ($n = 1419$) | FORWARDREJECT | 115 | 39 | 18 | 11 | 7 | 4 | 1 | 0 |
| | **Ours** | **106** | **37** | **18** | **11** | **7** | **4** | **1** | **0** |
| | Relative Improvement (%) | 7.83% | 5.13% | 0.00% | 0.00% | 0.00% | 0.00% | 0.00% | N/A |
| ICLR 2020 | ALLREJECT | 206 | 62 | 33 | 21 | 14 | 8 | 4 | 1 |
| ($n = 2213$) | FORWARDREJECT | 189 | 60 | 33 | 21 | 14 | 8 | 4 | 1 |
| | **Ours** | **177** | **56** | **29** | **18** | **11** | **7** | **4** | **1** |
| | Relative Improvement (%) | 6.35% | 6.67% | 12.12% | 14.29% | 21.43% | 12.50% | 0.00% | 0.00% |
| ICLR 2021 | ALLREJECT | 363 | 140 | 70 | 37 | 21 | 13 | 8 | 5 |
| ($n = 2594$) | FORWARDREJECT | 328 | 129 | 65 | 35 | 21 | 13 | 8 | 5 |
| | **Ours** | **303** | **120** | **65** | **35** | **21** | **13** | **8** | **5** |
| | Relative Improvement (%) | 7.62% | 6.98% | 0.00% | 0.00% | 0.00% | 0.00% | 0.00% | 0.00% |
| ICLR 2022 | ALLREJECT | 363 | 141 | 61 | 24 | 13 | 7 | 1 | 0 |
| ($n = 2617$) | FORWARDREJECT | 326 | 132 | 59 | 24 | 13 | 7 | 1 | 0 |
| | **Ours** | **296** | **124** | **56** | **23** | **13** | **7** | **1** | **0** |
| | Relative Improvement (%) | 9.20% | 6.06% | 5.08% | 4.17% | 0.00% | 0.00% | 0.00% | N/A |
| ICLR 2023 | ALLREJECT | 572 | 196 | 98 | 50 | 27 | 11 | 2 | 0 |
| ($n = 3793$) | FORWARDREJECT | 506 | 181 | 91 | 45 | 22 | 8 | 2 | 0 |
| | **Ours** | **460** | **166** | **84** | **43** | **20** | **8** | **2** | **0** |
| | Relative Improvement (%) | 9.09% | 8.29% | 7.69% | 4.44% | 9.09% | 0.00% | 0.00% | N/A |
| ICLR 2024 | ALLREJECT | 1797 | 811 | 384 | 186 | 104 | 58 | 30 | 16 |
| ($n = 7404$) | FORWARDREJECT | 1553 | 720 | 342 | 170 | 95 | 53 | 26 | 13 |
| | **Ours** | **1393** | **637** | **303** | **149** | **83** | **44** | **21** | **12** |
| | Relative Improvement (%) | 10.30% | 11.53% | 11.40% | 12.35% | 12.63% | 16.98% | 19.23% | 7.69% |
| ICLR 2025 | ALLREJECT | 3464 | 1807 | 995 | 554 | 294 | 158 | 89 | 51 |
| ($n = 11672$) | FORWARDREJECT | 2984 | 1577 | 889 | 499 | 273 | 151 | 83 | 47 |
| | **Ours** | **2668** | **1379** | **773** | **438** | **238** | **132** | **74** | **43** |
| | Relative Improvement (%) | 10.59% | 12.56% | 13.05% | 12.22% | 12.82% | 12.58% | 10.84% | 8.51% |

_Submission". For ICLR 2024–2025, we need to switch to OpenReview API v2 with the invitation link "`ICLR.cc/{year}/Conference/-/Submission`". Dataset statistics can be seen in Table 2.

Figure 2 plots per-author submission counts on a log scale. The counts follow a power-law distribution across different submission numbers. Owing to space limits, we show only ICLR 2023–2025 results here, and earlier years are provided in Appendix E.

Note that some papers may be missed by the OpenReview API, so our counts of papers and authors may not exactly match the official totals. However, this gap is small relative to public ICLR statistics and does not affect the validity of our empirical results.

**Baselines.** To the best of our knowledge, we are among the first to revisit submission-limit desk rejection, so our only baselines are the policies now used by major conferences, Algorithm 1 and Algorithm 2, which reject every paper with a non-compliant author in submission order. We omit Algorithm 5 because it is equivalent to Algorithm 2. These policies are compared against our method (Algorithm 4), which maximizes desk acceptance to improve author welfare.

**Reproducibility.** For the linear program solver LPSOLVER in Algorithm 4, we use the standard linear program solver in the Python PuLP library v3.2.1 [4]. The code is run on a server with Python 3.11 and Intel Xeon CPU with 2 vCPUs (virtual CPUs) and 13GB of RAM. No experiments have incorporated the use of GPUs.

Before running the solver, we speed up computation by removing all safe authors whose papers have no co-authors exceeding the submission limit; such papers face no risk of desk rejection. The exper-

---

[4]https://pypi.org/project/PuLP

iments are deterministic and contain no randomness, so we report single results without variances or p-values.

## 5.2 COMPARISON RESULTS

In this study, we compare our Algorithm 4 with the desk-rejection policies now policies in AI conferences, namely ALLREJECT (Algorithm 1) and FORWARDREJECT (Algorithm 2). We set the submission limit to $b \in \{4, 5, \cdots, 25\}$ and run each algorithm on every year of data. These values are practical, since some venues allow only a few submissions (e.g., HICSS[5] allows up to 5 and KDD 2025 allows 7), while others allow many (e.g., CVPR 2025 allows 25).

Here, we only report the results for submission limit $b \in \{4, 7, 10, 13, 16, 19, 22, 25\}$ using ICLR 2018–2025 data. This choice is reasonable since we consider both space limits and the small scale of the early years: ICLR 2013, 2014, and 2017 had few submissions (e.g., only 67 in 2013) and are less representative. Detailed results with all $b$ values and all years of ICLR data can be found in Appendix E.

**Main Findings.** All the comparison results are presented in Table 3. From the table, we have the following observations:

(i) The proposed desk-rejection method consistently outperforms the current desk-rejection policies used in existing conferences. Our improvement is dramatic, reaching up to $19.23\%$ relative improvement (see the $b = 22$ case for ICLR 2024). In cases where we do not observe relative improvement compared with the baselines, the submission limit $b$ is significantly larger than both the maximum and average number of submissions per author, so there are not enough papers requiring desk-rejection. This demonstrates the effectiveness of our proposed method in saving thousands of authors from having their papers desk-rejected.

(ii) As conference scale increases, the advantage of our method becomes more significant. For instance, in ICLR 2018, where the number of submissions was very small, our method only provides improvements in the $b = 4$ and $b = 7$ cases. In contrast, for later years such as ICLR 2024 and 2025, our method shows improvement across all $b$ values. In recent years, the scale of AI conferences has surged rapidly. This suggests a promising future for our method in supporting conferences with increasing submission volumes.

**Running Time.** Note that our algorithm is very efficient in practice, as all the numbers in Table 3 are computed within at most 53.64 seconds. This is already quite fast and efficient, so we are not focusing on runtime optimization in this work. We believe further improving efficiency could be an interesting future direction.

## 6 CONCLUSION

In this work, we present a pioneering study on improving author welfare in submission-limit desk-rejection systems. Specifically, we present an important formulation of the submission-limit-based desk-rejection, namely the maximum desk-acceptance problem, which is an integer program maximizing the number of papers that proceed to review while following per-author submission limits. Building on this formulation, we propose a two-stage solution that firstly solves the linear program relaxation of the original integer program, and then converts the fractional solution to a provably feasible integer solution via a specific rounding scheme. The resulting algorithm preserves the submission limit for every author while explicitly maximizing the number of papers that proceed to review. Our formulation and algorithm offer an elegant replacement for current desk-rejection rules, leading to a desirable balance between reviewer workload and authors' opportunity to contribute. Future work could integrate our allocation step with modern reviewer-assignment systems, creating an end-to-end pipeline that has direct transformative social impact in the real world.

---

[5]https://hicss.hawaii.edu/authors/

## ETHICS STATEMENT

Our work aims to minimize unnecessary desk rejections in AI conferences, particularly those arising from submission-limit policies. By seeking solutions that reject as few papers as possible while following conference submission constraints, we promote author welfare, especially for early-career researchers who may be disproportionately affected, since a desk rejection may have little impact on well-established collaborators with many submissions but can be critical for those with fewer papers. In doing so, we contribute to a more inclusive and equitable research community, giving early-career researchers more opportunities to submit their work.

## REPRODUCIBILITY STATEMENT

For theoretical reproducibility, all assumptions are explicitly stated in the theorems and lemmas. Complete proofs are provided in Appendix B, and each main-text statement includes a reference to its corresponding proof.

For empirical reproducibility, due to copyright constraints, we may not release code and data at this stage. However, all code and data will be made publicly available upon paper acceptance. Section 5 describes how to crawl the ICLR datasets using the OpenReview API, our parameter settings, and how to solve the proposed optimization problem with standard linear programming solvers. Since all algorithms in this paper include pseudocode, and our main method (Algorithm 4) is highly simple, we do not foresee empirical reproducibility issues.

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

# Appendix

**Roadmap.** In Section A, we discuss additional related works for this paper. In Section B, we provide the missing proofs. In Section C, we supplement the missing backward desk-rejection algorithm for Section 3. In Section D, we show the source of conference submission limit policies. In Section E, we present the additional experiments.

## A    ADDITIONAL RELATED WORKS

**Desk-Rejection Mechanisms.** A variety of desk-rejection mechanisms have been developed to reduce the manual workload associated with the peer-review process (Ansell & Samuels, 2021). One of the most widely adopted rules is the rejection of submissions that violate anonymity requirements (Jefferson et al., 2002; Tennant, 2018), which is essential for ensuring unbiased reviews across institutions and career stages, and for mitigating conflicts of interest. Another common policy targets duplicate and dual submissions (Stone, 2003; Leopold, 2013), helping to avoid redundant reviewer effort and maintain publication ethics. Plagiarism (King & ChatGPT, 2023; Elali & Rachid, 2023) also remains a key reason for desk rejection, as it compromises academic integrity, infringes on intellectual property, and undermines the credibility of research contributions. In response to the sharp increase in AI conference submissions, newer forms of desk-rejection policies have recently emerged (Leyton-Brown et al., 2024). For instance, IJCAI 2020 and NeurIPS 2020 adopted a rapid rejection strategy, allowing area chairs to desk-reject papers based on a brief review of the abstract and main content. While this method aimed to alleviate reviewer load, it introduced instability and might lead to the rejection of potentially promising work. As a result, it has seen limited adoption compared to more systematic approaches, such as author-level submission limits (see Table 4 for major conferences that adopt or do not adopt this policy), which are the primary focus of this paper. To the best of our knowledge, there is limited prior work that formally studies submission-limit-based desk-rejection mechanisms. Our work is among the first to provide a rigorous formulation and optimization-based analysis of this emerging policy class.

**Linear Programming and Semidefinite Programming.** Linear programming is a foundational topic in computer science and optimization. The Simplex algorithm, introduced by Dantzig (Dantzig, 1951), is a cornerstone of linear programming, though it has exponential worst-case complexity. The Ellipsoid method provides a polynomial-time guarantee (Khachiyan, 1980), marking a significant theoretical breakthrough, but it is often outperformed by the Simplex method in practice. A major advancement came with the introduction of the interior-point method (Karmarkar, 1984), which combines polynomial-time guarantees with strong empirical performance across a broad range of real-world problems. This breakthrough initiated a rich line of research aimed at accelerating interior-point algorithms for solving classical optimization problems. The impact of interior-point methods extends beyond linear programming to a variety of complex tasks (Vaidya, 1987; Renegar, 1988; Vaidya, 1989; Daitch & Spielman, 2008; Lee & Sidford, 2013; 2014; 2019; Song, 2019; Cohen et al., 2021a; Lee et al., 2019; Brand, 2020; Brand et al., 2020; Jiang et al., 2021; Song & Yu, 2021; Gu & Song, 2022). Furthermore, both interior-point methods are central to solving semidefinite programming (SDP) problems (Song, 2019; Jiang et al., 2020a; Song et al., 2023; Gu & Song, 2022; Huang et al., 2022a;b). Linear programming and semidefinite programming are also widely applied in machine learning theory, especially in areas such as empirical risk minimization (Lee et al., 2019; Song et al., 2022; Qin et al., 2023) and support vector machines (Gu et al., 2023; Gao et al., 2023).

**Integer Programming.** Integer Programming (IP), and even the special case of Integer Linear Programming (ILP), are computationally hard in general. Solving ILP exactly is NP-complete, and there are no known strongly polynomial-time algorithms for general instances. However, several structured classes of ILP admit efficient algorithms. For instance, ILPs defined by totally unimodular constraint matrices can be solved in strongly polynomial time via linear programming relaxations (Hochbaum & Shanthikumar, 1990). Beyond this, other tractable cases include bimodular ILPs (Artmann et al., 2017), binet matrices (Appa et al., 2007), and $n$-fold integer programs with constant block dimensions, which admit fixed-parameter tractable algorithms (Hemmecke et al., 2013; De Loera et al., 2015). These tractable subfamilies have found applications in scheduling (Ryan & Foster, 1981), network flows (Bazaraa et al., 2011), and multicommodity routing problems (Ramkumar et al., 2012). Despite these advances, practical integer programming solvers

such as CPLEX (Cplex, 2009) and Gurobi (Gurobi Optimization, LLC, 2024) rely on branch-and-bound (Lawler & Wood, 1966; Morrison et al., 2016), cutting planes (Kelley, 1960; Jiang et al., 2020b), and heuristics (Lenstra Jr, 1983), which do not guarantee polynomial time in the worst case. Recent research also explores approximation algorithms and relaxation hierarchies for general IPs (Barak et al., 2014), as well as learning-augmented solvers (Khalil et al., 2016) that integrate machine learning to guide search and branching decisions. In our work, we formulate the submission limit problem as an integer program and solve its relaxed linear program to obtain approximate solutions.

## B  MISSING PROOFS

In Section B.1, we show the missing proofs for Section 3. In Section B.2, we present the missing proofs for Section 4.

### B.1  MISSING PROOFS IN SECTION 3

**Proposition B.1** (Correctness of ALLREJECT, formal version of Proposition 3.5). *In $O(nk_1)$ time, Algorithm 1 produces a feasible solution for $x$ as defined in Definition 3.1.*

*Proof.* Let $k_1$ and $k_2$ defined as Definition 3.4.

Since the algorithm iterates through all the $n$ papers and each iteration $i \in [n]$ ends in $O(k_1)$ time, the algorithm ends in $O(nk_1)$ time. For each author $i \in [n]$, line 6 ensures that the author cannot have more than $b$ papers, which matches Definition 3.1.

Thus, we complete the proof. □

**Proposition B.2** (Correctness of FORWARDREJECT, formal version of Proposition 3.6). *Algorithm 2 takes $O(mk_2)$ time and outputs the feasible solution for $x$ as defined in Definition 3.1.*

*Proof.* **Part 1. Correctness.** Considering the main loops in lines 3-10, we have the following:

- Before running the loop (i.e., $j = 0$), the submission limit of all authors trivially holds.

- In each iteration, the paper acceptance will not break the submission limit for its authors, since we have line 5. This means that, if at iteration $j$ the submission limit of all authors is satisfied, then at iteration $(j + 1)$ the submission limit of all authors will still hold.

Thus, by induction, we can conclude that the result produced by Algorithm 2 matches Definition 3.1.

**Part 2. Stop in $O(mk_2)$ time.** Since the algorithm iterates over all the $m$ papers and in each iteration $j \in [m]$, the time complexity is $O(k_2)$, we can conclude that the algorithm ends in $O(mk_2)$ time.

Thus, we finish the proof. □

### B.2  MISSING PROOFS IN SECTION 4

**Theorem B.3** (Correctness of MAXROUNDING, formal version of Theorem 4.6). *Algorithm 3 takes $O(nk_1 + mk_1k_2)$ time and outputs the feasible solution for $x$ as defined in Definition 3.1.*

*Proof.* **Part 1. Correctness.** The while condition in line 7 ensures that the algorithm will not stop unless there are no fractional solutions. Thus, the constraint $x \in \{0, 1\}^m$ is satisfied. Inside the while loop (lines 7-19), all the authors affected by up rounding paper $l$ will be desk-rejected a sufficient number of papers, which ensures that the submission limit is satisfied. Therefore, we can conclude that Algorithm 3 produces a correct feasible solution.

**Part 2. Stop in $O(nk_1 + mk_1k_2)$ Time.**

Before running the while loop (lines 7-19), we first compute the set of fractional solutions $S$ in $O(m)$ time, and construct $T_i$ for all $i \in [n]$ in $O(nk_1)$ time. The while loop includes at most $m$

iterations, and in each iteration, we run a for loop with $O(k_1 k_2)$ time. This results in a total running time of $O(mk_1 k_2)$. Combining the time cost before the while loop and in the while loop, we can conclude that the algorithm stops in $O(nk_1 + mk_1 k_2)$ time.

Thus, we finish the proof.

$\square$

## C  THE MISSING BACKWARD ALGORITHM FOR SECTION 3

In this section, we present an alternative implementation of the conventional submission-limit-based desk-rejection algorithm and formally prove its correctness and time complexity. This supplements the previously omitted algorithm in Section 3.2.

---
**Algorithm 5** Current desk-reject algorithm that rejects extra papers sequentially

---
1: **procedure** BACKWARDREJECT($A \in \{0,1\}^{n \times m}, n, m, b \in \mathbb{N}_+$)
2:     **for** $i \in [n]$ **do**                                         ▷ This loop takes $O(nk_1)$ time.
3:         $P_i \leftarrow \{j : j \in [m], A_{i,j} = 1\}, \forall i \in [n]$ ▷ The set of desk-accept paper for each author $i$.
4:     **end for**
5:     **for** $j = m \dots 1$ **do**
6:         $S \leftarrow \{i : i \in [n], A_{i,j} = 1\}$         ▷ All the authors for paper $j$. This take $O(k_2)$ time.
7:         **if** exists $i \in S$ such that $|P_i| > b$ **then**     ▷ We desk-reject a paper if it has at least one author that exceeds the submission limit. Check this takes $O(k_2)$ time.
8:             **for** $i \in S$ **do**
9:                 $P_i \leftarrow P_i \setminus \{j\}$
10:             **end for**
11:         **end if**
12:     **end for**
13:     $P \leftarrow \bigcup_{i=1}^{n} P_i$                                 ▷ All the accepted papers
14:     $x \leftarrow \mathbf{0}_m$                                     ▷ The desk-rejection result
15:     **for** $j \in P$ **do**
16:         $x_j \leftarrow 1$
17:     **end for**
18:     **return** $x$                                   ▷ $x \in \{0,1\}^m$
19: **end procedure**

---

**Proposition C.1** (Correctness of BACKWARDREJECT). *Algorithm 5 takes $O(nk_1 + mk_2)$ time and outputs the feasible solution for $x$ as defined in Definition 3.1.*

*Proof.* **Part 1. Correctness.** We can prove this by contradiction. Specifically, we first assume that after running Algorithm 5, there exists an author $i \in [n]$ that still exceeds the submission limit. In this case, the final $P_i$ should satisfy $|P_i| > 0$. Otherwise, the author has no papers and cannot exceed any valid submission limit.

After checking all the $m$ papers, the author $i$ still exceeds the limit, Algorithm 3 must have rejected all the papers that include $i$ as an author in the for loop. Thus, the final paper set of $i$ satisfies $|P_i| = 0$. This contradicts the fact that $|P_i| > 0$.

Therefore, we have shown that there is no author who still exceeds the submission limit after running Algorithm 5.

**Part 2. Stop in $O(nk_1 + mk_2)$ time.** First, computing $P_i$ for all $i \in [n]$ takes $O(k_1)$ time. Then, we iterate over all the $m$ papers and each iteration $j \in [m]$ takes $O(k_2)$ time, resulting in $O(mk_2)$ time in total. Combining two parts of the algorithm yields the $O(nk_1 + mk_2)$ time complexity.

Thus, we complete the proof. $\square$

Table 4: Summary of 13 major AI conferences, showing founding years, current editions (as of 2025), the existence of submission limits, and the frequency of previous venues. The current edition of ECCV is 2024, since it does not occur in 2025. Most conferences, such as NeurIPS, ICLR, ACL, EMNLP, KDD, and WSDM, are held annually, while ECCV has been held biennially. CVPR has been held annually since 1985, except for 1986, 1987, 1990, and 1995. ICCV was held irregularly in 1987, 1988, 1990, 1993, 1995, and 1998, and has followed a biennial schedule since 1999. ICML was held irregularly in 1980, 1983, and 1985, and annually since 1988. NAACL was held irregularly in 2000, 2001, 2003, 2004, 2006, 2007, 2009, 2010, 2012, 2013, 2015, 2016, 2018, 2019, 2021, 2022, 2024, and 2025. IJCAI was held biennially from 1969 to 2015, and annually since 2016. AAAI has been held annually since 1980, except for 1981, 1985, 1989, 1995, 2001, 2003, and 2009.

| Conference | Founded | Edition | Limit | Frequency |
|---|---|---|---|---|
| CVPR | 1984 | 38th, 2025 | Yes | Annual (since 1985; skipped 1986, 1987, 1990, 1995) |
| ICCV | 1987 | 20th, 2025 | Yes | Irregular until 1999; annual thereafter |
| ECCV | 1990 | 18th, 2024 | No | Biennial (since 1990) |
| ICML | 1980 | 42nd, 2025 | No | Irregular until 1988; annual thereafter |
| NeurIPS | 1987 | 39th, 2025 | No | Annual (since 1987) |
| ICLR | 2013 | 13th, 2025 | No | Annual (since 2013) |
| ACL | 1963 | 63rd, 2025 | No | Annual (since 1962) |
| EMNLP | 1996 | 30th, 2025 | No | Annual (since 1996) |
| NAACL | 2000 | 18th, 2025 | No | Irregular |
| KDD | 1995 | 31st, 2025 | Yes | Annual (since 1995) |
| WSDM | 2008 | 18th, 2025 | Yes | Annual (since 2008) |
| IJCAI | 1969 | 34th, 2025 | Yes | Biennial until 2015; annual thereafter |
| AAAI | 1980 | 39th, 2025 | Yes | Annual (since 1980; skipped 1981, 1985, 1989, 1995, 2001, 2003, 2009) |

## D  CONFERENCE SUBMISSION LIMIT POLICIES

In the introduction, Table 1 provides only a high-level overview of each venue's year and per-author submission limit. Here, we supply the complete set of author-guideline URLs for each conference and year. Specifically, we cover: CVPR (2024-2025) (Committees, 2024b; 2025b), ICCV (2023, 2025) (Committees, 2023b; 2025c), AAAI (2022-2025) (Committees, 2022a; 2023a; 2024a; 2025a), WSDM (2020-2025) (Committees, 2020c; 2021b; 2022c; 2023e; 2024f; 2025g), IJCAI (2017-2025) (Committees, 2017; 2018; 2019; 2020a; 2021a; 2022b; 2023c; 2024d; 2025e), KDD (2023-2025) (Committees, 2023d; 2024e; 2025f), and ICDE (2024-2025) (Committees, 2024c; 2025d).

We summarize major AI conferences in Table 4, including their founding years and whether they enforce submission limits. From the table, we observe that 6 out of 13 major AI conferences employ submission-limit-based desk-rejection policies, indicating their growing prevalence. This highlights the potential impact of our work, which applies to a wide range of conferences and can improve author welfare across the whole AI community.

## E  ADDITIONAL EXPERIMENTS

We present the additional submission frequency bar charts in Figures 3– 6, and the detailed comparison results with all years of data and submission limit levels can be found in Tables 5– 8. From the tables, we observe that, despite some extreme cases where the limit $b$ is too large and few papers require desk-rejection, our proposed method consistently outperforms all baselines, with particularly strong results for ICLR 2023 to ICLR 2025.

Table 5: Number of rejected papers for conventional desk rejection and our proposed desk rejection methods at submission limits $b = 1$ through $b = 8$. Lower values indicate fewer papers desk-rejected.

| Dataset | Method | $b=1$ | $b=2$ | $b=3$ | $b=4$ | $b=5$ | $b=6$ | $b=7$ | $b=8$ |
|---|---|---|---|---|---|---|---|---|---|
| ICLR 2013 | ALLREJECT | 19 | 9 | 7 | 5 | 3 | 1 | 0 | 0 |
| ($n=67$) | FORWARDREJECT | 19 | 9 | 7 | 5 | 3 | 1 | 0 | 0 |
| | **Ours** | 19 | 9 | 7 | 5 | 3 | 1 | 0 | 0 |
| | Relative Improvement (%) | 0.00% | 0.00% | 0.00% | 0.00% | 0.00% | 0.00% | 0.00% | 0.00% |
| ICLR 2014 | ALLREJECT | 16 | 8 | 5 | 3 | 2 | 1 | 0 | 0 |
| ($n=69$) | FORWARDREJECT | 16 | 7 | 5 | 3 | 2 | 1 | 0 | 0 |
| | **Ours** | 16 | 7 | 5 | 3 | 2 | 1 | 0 | 0 |
| | Relative Improvement (%) | 0.00% | 0.00% | 0.00% | 0.00% | 0.00% | 0.00% | 0.00% | 0.00% |
| ICLR 2017 | ALLREJECT | 186 | 87 | 52 | 24 | 11 | 4 | 1 | 0 |
| ($n=490$) | FORWARDREJECT | 173 | 83 | 42 | 22 | 11 | 4 | 1 | 0 |
| | **Ours** | 163 | 76 | 41 | 21 | 10 | 4 | 1 | 0 |
| | Relative Improvement (%) | 5.78% | 8.43% | 2.38% | 4.55% | 9.09% | 0.00% | 0.00% | 0.00% |
| ICLR 2018 | ALLREJECT | 375 | 177 | 95 | 56 | 36 | 25 | 18 | 11 |
| ($n=935$) | FORWARDREJECT | 336 | 157 | 85 | 53 | 36 | 25 | 18 | 11 |
| | **Ours** | 313 | 143 | 78 | 51 | 35 | 24 | 17 | 11 |
| | Relative Improvement (%) | 6.85% | 8.92% | 8.24% | 3.77% | 2.78% | 4.00% | 5.56% | 0.00% |
| ICLR 2019 | ALLREJECT | 629 | 332 | 196 | 127 | 89 | 62 | 43 | 31 |
| ($n=1419$) | FORWARDREJECT | 568 | 293 | 175 | 115 | 80 | 56 | 39 | 28 |
| | **Ours** | 536 | 270 | 160 | 106 | 73 | 52 | 37 | 28 |
| | Relative Improvement (%) | 5.63% | 7.85% | 8.57% | 7.83% | 8.75% | 7.14% | 5.13% | 0.00% |
| ICLR 2020 | ALLREJECT | 1103 | 580 | 335 | 206 | 137 | 95 | 62 | 47 |
| ($n=2213$) | FORWARDREJECT | 976 | 515 | 305 | 189 | 129 | 89 | 60 | 46 |
| | **Ours** | 920 | 476 | 283 | 177 | 120 | 81 | 56 | 42 |
| | Relative Improvement (%) | 5.74% | 7.57% | 7.21% | 6.35% | 6.98% | 8.99% | 6.67% | 8.70% |
| ICLR 2021 | ALLREJECT | 1381 | 809 | 516 | 363 | 261 | 192 | 140 | 103 |
| ($n=2594$) | FORWARDREJECT | 1240 | 721 | 461 | 328 | 233 | 173 | 129 | 97 |
| | **Ours** | 1168 | 668 | 431 | 303 | 212 | 159 | 120 | 93 |
| | Relative Improvement (%) | 5.81% | 7.35% | 6.51% | 7.62% | 9.01% | 8.09% | 6.98% | 4.12% |
| ICLR 2022 | ALLREJECT | 1438 | 834 | 539 | 363 | 257 | 187 | 141 | 108 |
| ($n=2617$) | FORWARDREJECT | 1285 | 746 | 484 | 326 | 235 | 174 | 132 | 100 |
| | **Ours** | 1203 | 686 | 445 | 296 | 211 | 161 | 124 | 94 |
| | Relative Improvement (%) | 6.38% | 8.04% | 8.06% | 9.20% | 10.21% | 7.47% | 6.06% | 6.00% |
| ICLR 2023 | ALLREJECT | 2214 | 1343 | 848 | 572 | 397 | 279 | 196 | 150 |
| ($n=3793$) | FORWARDREJECT | 1952 | 1163 | 738 | 506 | 352 | 251 | 181 | 141 |
| | **Ours** | 1842 | 1085 | 686 | 460 | 321 | 228 | 166 | 131 |
| | Relative Improvement (%) | 5.64% | 6.71% | 7.05% | 9.09% | 8.81% | 9.16% | 8.29% | 7.09% |
| ICLR 2024 | ALLREJECT | 4883 | 3307 | 2409 | 1797 | 1358 | 1036 | 811 | 623 |
| ($n=7404$) | FORWARDREJECT | 4328 | 2872 | 2071 | 1553 | 1185 | 913 | 720 | 554 |
| | **Ours** | 4094 | 2636 | 1872 | 1393 | 1061 | 816 | 637 | 492 |
| | Relative Improvement (%) | 5.41% | 8.22% | 9.61% | 10.30% | 10.46% | 10.62% | 11.53% | 11.19% |
| ICLR 2025 | ALLREJECT | 8200 | 5945 | 4482 | 3464 | 2760 | 2222 | 1807 | 1476 |
| ($n=11672$) | FORWARDREJECT | 7274 | 5143 | 3854 | 2984 | 2377 | 1920 | 1577 | 1299 |
| | **Ours** | 6890 | 4720 | 3472 | 2668 | 2100 | 1687 | 1379 | 1134 |
| | Relative Improvement (%) | 5.28% | 8.22% | 9.91% | 10.59% | 11.65% | 12.14% | 12.56% | 12.70% |

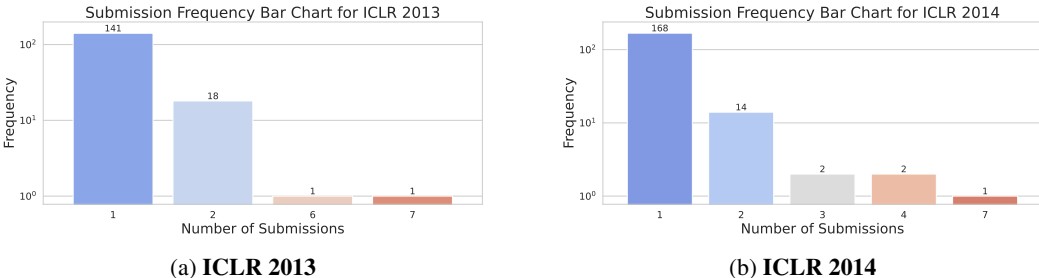

(a) **ICLR 2013**        (b) **ICLR 2014**

Figure 3: **Submission Frequency Bar Charts for ICLR 2013–2014.**

Table 6: Number of rejected papers for conventional desk rejection and our proposed desk rejection methods at submission limits $b = 9$ through $b = 16$. Lower values indicate fewer papers desk-rejected.

| Dataset | Method | $b=9$ | $b=10$ | $b=11$ | $b=12$ | $b=13$ | $b=14$ | $b=15$ | $b=16$ |
|---|---|---|---|---|---|---|---|---|---|
| ICLR 2013 | ALLREJECT | 0 | 0 | 0 | 0 | 0 | 0 | 0 | 0 |
| ($n = 67$) | FORWARDREJECT | 0 | 0 | 0 | 0 | 0 | 0 | 0 | 0 |
| | **Ours** | 0 | 0 | 0 | 0 | 0 | 0 | 0 | 0 |
| | Relative Improvement (%) | 0.00% | 0.00% | 0.00% | 0.00% | 0.00% | 0.00% | 0.00% | 0.00% |
| ICLR 2014 | ALLREJECT | 0 | 0 | 0 | 0 | 0 | 0 | 0 | 0 |
| ($n = 69$) | FORWARDREJECT | 0 | 0 | 0 | 0 | 0 | 0 | 0 | 0 |
| | **Ours** | 0 | 0 | 0 | 0 | 0 | 0 | 0 | 0 |
| | Relative Improvement (%) | 0.00% | 0.00% | 0.00% | 0.00% | 0.00% | 0.00% | 0.00% | 0.00% |
| ICLR 2017 | ALLREJECT | 0 | 0 | 0 | 0 | 0 | 0 | 0 | 0 |
| ($n = 490$) | FORWARDREJECT | 0 | 0 | 0 | 0 | 0 | 0 | 0 | 0 |
| | **Ours** | 0 | 0 | 0 | 0 | 0 | 0 | 0 | 0 |
| | Relative Improvement (%) | 0.00% | 0.00% | 0.00% | 0.00% | 0.00% | 0.00% | 0.00% | 0.00% |
| ICLR 2018 | ALLREJECT | 8 | 5 | 2 | 0 | 0 | 0 | 0 | 0 |
| ($n = 935$) | FORWARDREJECT | 8 | 5 | 2 | 0 | 0 | 0 | 0 | 0 |
| | **Ours** | 8 | 5 | 2 | 0 | 0 | 0 | 0 | 0 |
| | Relative Improvement (%) | 0.00% | 0.00% | 0.00% | 0.00% | 0.00% | 0.00% | 0.00% | 0.00% |
| ICLR 2019 | ALLREJECT | 22 | 18 | 15 | 13 | 11 | 9 | 8 | 7 |
| ($n = 1419$) | FORWARDREJECT | 22 | 18 | 15 | 13 | 11 | 9 | 8 | 7 |
| | **Ours** | 22 | 18 | 15 | 13 | 11 | 9 | 8 | 7 |
| | Relative Improvement (%) | 0.00% | 0.00% | 0.00% | 0.00% | 0.00% | 0.00% | 0.00% | 0.00% |
| ICLR 2020 | ALLREJECT | 38 | 33 | 29 | 25 | 21 | 18 | 16 | 14 |
| ($n = 2213$) | FORWARDREJECT | 38 | 33 | 28 | 24 | 21 | 18 | 16 | 14 |
| | **Ours** | 34 | 29 | 25 | 21 | 18 | 15 | 13 | 11 |
| | Relative Improvement (%) | 10.53% | 12.12% | 10.71% | 12.50% | 14.29% | 16.67% | 18.75% | 21.43% |
| ICLR 2021 | ALLREJECT | 85 | 70 | 59 | 47 | 37 | 30 | 25 | 21 |
| ($n = 2594$) | FORWARDREJECT | 79 | 65 | 55 | 44 | 35 | 29 | 25 | 21 |
| | **Ours** | 77 | 65 | 54 | 43 | 35 | 29 | 25 | 21 |
| | Relative Improvement (%) | 2.53% | 0.00% | 1.82% | 2.27% | 0.00% | 0.00% | 0.00% | 0.00% |
| ICLR 2022 | ALLREJECT | 79 | 61 | 46 | 35 | 24 | 18 | 15 | 13 |
| ($n = 2617$) | FORWARDREJECT | 76 | 59 | 44 | 32 | 24 | 18 | 15 | 13 |
| | **Ours** | 72 | 56 | 42 | 31 | 23 | 18 | 15 | 13 |
| | Relative Improvement (%) | 5.26% | 5.08% | 4.55% | 3.12% | 4.17% | 0.00% | 0.00% | 0.00% |
| ICLR 2023 | ALLREJECT | 119 | 98 | 81 | 65 | 50 | 42 | 34 | 27 |
| ($n = 3793$) | FORWARDREJECT | 113 | 91 | 75 | 59 | 45 | 37 | 29 | 22 |
| | **Ours** | 104 | 84 | 69 | 55 | 43 | 34 | 27 | 20 |
| | Relative Improvement (%) | 7.96% | 7.69% | 8.00% | 6.78% | 4.44% | 8.11% | 6.90% | 9.09% |
| ICLR 2024 | ALLREJECT | 486 | 384 | 303 | 239 | 186 | 151 | 123 | 104 |
| ($n = 7404$) | FORWARDREJECT | 435 | 342 | 274 | 215 | 170 | 139 | 115 | 95 |
| | **Ours** | 383 | 303 | 239 | 188 | 149 | 122 | 99 | 83 |
| | Relative Improvement (%) | 11.95% | 11.40% | 12.77% | 12.56% | 12.35% | 12.23% | 13.91% | 12.63% |
| ICLR 2025 | ALLREJECT | 1207 | 995 | 822 | 681 | 554 | 456 | 368 | 294 |
| ($n = 11672$) | FORWARDREJECT | 1070 | 889 | 740 | 614 | 499 | 409 | 337 | 273 |
| | **Ours** | 935 | 773 | 640 | 533 | 438 | 360 | 294 | 238 |
| | Relative Improvement (%) | 12.62% | 13.05% | 13.51% | 13.19% | 12.22% | 11.98% | 12.76% | 12.82% |

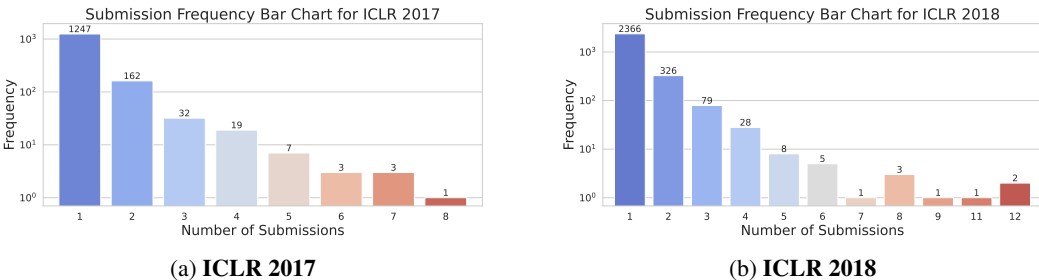

(a) **ICLR 2017**  (b) **ICLR 2018**

Figure 4: **Submission Frequency Bar Charts for ICLR 2017–2018.**

Table 7: Number of rejected papers for conventional desk rejection and our proposed desk rejection methods at submission limits $b = 17$ through $b = 24$. Lower values indicate fewer papers desk-rejected.

| Dataset | Method | $b=17$ | $b=18$ | $b=19$ | $b=20$ | $b=21$ | $b=22$ | $b=23$ | $b=24$ |
|---------|--------|--------|--------|--------|--------|--------|--------|--------|--------|
| ICLR 2013 | ALLREJECT | 0 | 0 | 0 | 0 | 0 | 0 | 0 | 0 |
| ($n=67$) | FORWARDREJECT | 0 | 0 | 0 | 0 | 0 | 0 | 0 | 0 |
| | **Ours** | 0 | 0 | 0 | 0 | 0 | 0 | 0 | 0 |
| | Relative Improvement (%) | 0.00% | 0.00% | 0.00% | 0.00% | 0.00% | 0.00% | 0.00% | 0.00% |
| ICLR 2014 | ALLREJECT | 0 | 0 | 0 | 0 | 0 | 0 | 0 | 0 |
| ($n=69$) | FORWARDREJECT | 0 | 0 | 0 | 0 | 0 | 0 | 0 | 0 |
| | **Ours** | 0 | 0 | 0 | 0 | 0 | 0 | 0 | 0 |
| | Relative Improvement (%) | 0.00% | 0.00% | 0.00% | 0.00% | 0.00% | 0.00% | 0.00% | 0.00% |
| ICLR 2017 | ALLREJECT | 0 | 0 | 0 | 0 | 0 | 0 | 0 | 0 |
| ($n=490$) | FORWARDREJECT | 0 | 0 | 0 | 0 | 0 | 0 | 0 | 0 |
| | **Ours** | 0 | 0 | 0 | 0 | 0 | 0 | 0 | 0 |
| | Relative Improvement (%) | 0.00% | 0.00% | 0.00% | 0.00% | 0.00% | 0.00% | 0.00% | 0.00% |
| ICLR 2018 | ALLREJECT | 0 | 0 | 0 | 0 | 0 | 0 | 0 | 0 |
| ($n=935$) | FORWARDREJECT | 0 | 0 | 0 | 0 | 0 | 0 | 0 | 0 |
| | **Ours** | 0 | 0 | 0 | 0 | 0 | 0 | 0 | 0 |
| | Relative Improvement (%) | 0.00% | 0.00% | 0.00% | 0.00% | 0.00% | 0.00% | 0.00% | 0.00% |
| ICLR 2019 | ALLREJECT | 6 | 5 | 4 | 3 | 2 | 1 | 0 | 0 |
| ($n=1419$) | FORWARDREJECT | 6 | 5 | 4 | 3 | 2 | 1 | 0 | 0 |
| | **Ours** | 6 | 5 | 4 | 3 | 2 | 1 | 0 | 0 |
| | Relative Improvement (%) | 0.00% | 0.00% | 0.00% | 0.00% | 0.00% | 0.00% | 0.00% | 0.00% |
| ICLR 2020 | ALLREJECT | 12 | 10 | 8 | 6 | 5 | 4 | 3 | 2 |
| ($n=2213$) | FORWARDREJECT | 12 | 10 | 8 | 6 | 5 | 4 | 3 | 2 |
| | **Ours** | 9 | 8 | 7 | 6 | 5 | 4 | 3 | 2 |
| | Relative Improvement (%) | 25.00% | 20.00% | 12.50% | 0.00% | 0.00% | 0.00% | 0.00% | 0.00% |
| ICLR 2021 | ALLREJECT | 18 | 15 | 13 | 11 | 9 | 8 | 7 | 6 |
| ($n=2594$) | FORWARDREJECT | 18 | 15 | 13 | 11 | 9 | 8 | 7 | 6 |
| | **Ours** | 18 | 15 | 13 | 11 | 9 | 8 | 7 | 6 |
| | Relative Improvement (%) | 0.00% | 0.00% | 0.00% | 0.00% | 0.00% | 0.00% | 0.00% | 0.00% |
| ICLR 2022 | ALLREJECT | 11 | 9 | 7 | 5 | 3 | 1 | 0 | 0 |
| ($n=2617$) | FORWARDREJECT | 11 | 9 | 7 | 5 | 3 | 1 | 0 | 0 |
| | **Ours** | 11 | 9 | 7 | 5 | 3 | 1 | 0 | 0 |
| | Relative Improvement (%) | 0.00% | 0.00% | 0.00% | 0.00% | 0.00% | 0.00% | 0.00% | 0.00% |
| ICLR 2023 | ALLREJECT | 20 | 16 | 11 | 6 | 3 | 2 | 1 | 0 |
| ($n=3793$) | FORWARDREJECT | 16 | 12 | 8 | 5 | 3 | 2 | 1 | 0 |
| | **Ours** | 14 | 11 | 8 | 5 | 3 | 2 | 1 | 0 |
| | Relative Improvement (%) | 12.50% | 8.33% | 0.00% | 0.00% | 0.00% | 0.00% | 0.00% | 0.00% |
| ICLR 2024 | ALLREJECT | 88 | 71 | 58 | 47 | 38 | 30 | 25 | 20 |
| ($n=7404$) | FORWARDREJECT | 79 | 65 | 53 | 42 | 33 | 26 | 21 | 17 |
| | **Ours** | 69 | 55 | 44 | 34 | 27 | 21 | 17 | 14 |
| | Relative Improvement (%) | 12.66% | 15.38% | 16.98% | 19.05% | 18.18% | 19.23% | 19.05% | 17.65% |
| ICLR 2025 | ALLREJECT | 233 | 188 | 158 | 132 | 109 | 89 | 72 | 60 |
| ($n=11672$) | FORWARDREJECT | 218 | 178 | 151 | 126 | 103 | 83 | 66 | 55 |
| | **Ours** | 193 | 158 | 132 | 111 | 91 | 74 | 60 | 50 |
| | Relative Improvement (%) | 11.47% | 11.24% | 12.58% | 11.90% | 11.65% | 10.84% | 9.09% | 9.09% |

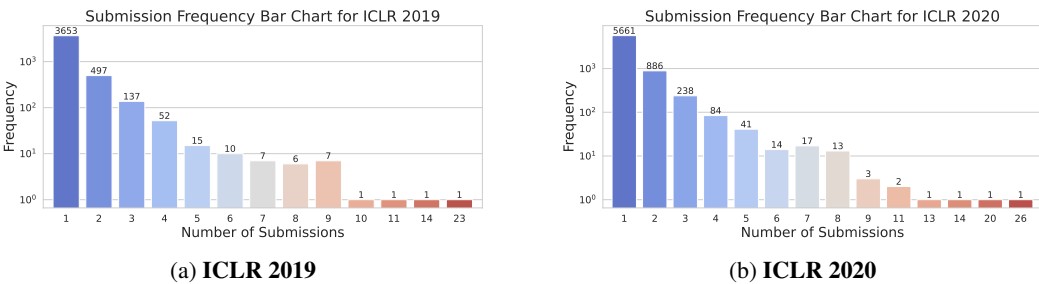

(a) **ICLR 2019**

(b) **ICLR 2020**

Figure 5: **Submission Frequency Bar Charts for ICLR 2019–2020.**

Table 8: Number of rejected papers for conventional desk rejection and our proposed desk rejection methods at submission limits $b = 25$ through $b = 32$. Lower values indicate fewer papers desk-rejected.

| Dataset | Method | $b=25$ | $b=26$ | $b=27$ | $b=28$ | $b=29$ | $b=30$ | $b=31$ | $b=32$ |
|---|---|---|---|---|---|---|---|---|---|
| ICLR 2013 | ALLREJECT | 0 | 0 | 0 | 0 | 0 | 0 | 0 | 0 |
| ($n=67$) | FORWARDREJECT | 0 | 0 | 0 | 0 | 0 | 0 | 0 | 0 |
| | **Ours** | 0 | 0 | 0 | 0 | 0 | 0 | 0 | 0 |
| | Relative Improvement (%) | 0.00% | 0.00% | 0.00% | 0.00% | 0.00% | 0.00% | 0.00% | 0.00% |
| ICLR 2014 | ALLREJECT | 0 | 0 | 0 | 0 | 0 | 0 | 0 | 0 |
| ($n=69$) | FORWARDREJECT | 0 | 0 | 0 | 0 | 0 | 0 | 0 | 0 |
| | **Ours** | 0 | 0 | 0 | 0 | 0 | 0 | 0 | 0 |
| | Relative Improvement (%) | 0.00% | 0.00% | 0.00% | 0.00% | 0.00% | 0.00% | 0.00% | 0.00% |
| ICLR 2017 | ALLREJECT | 0 | 0 | 0 | 0 | 0 | 0 | 0 | 0 |
| ($n=490$) | FORWARDREJECT | 0 | 0 | 0 | 0 | 0 | 0 | 0 | 0 |
| | **Ours** | 0 | 0 | 0 | 0 | 0 | 0 | 0 | 0 |
| | Relative Improvement (%) | 0.00% | 0.00% | 0.00% | 0.00% | 0.00% | 0.00% | 0.00% | 0.00% |
| ICLR 2018 | ALLREJECT | 0 | 0 | 0 | 0 | 0 | 0 | 0 | 0 |
| ($n=935$) | FORWARDREJECT | 0 | 0 | 0 | 0 | 0 | 0 | 0 | 0 |
| | **Ours** | 0 | 0 | 0 | 0 | 0 | 0 | 0 | 0 |
| | Relative Improvement (%) | 0.00% | 0.00% | 0.00% | 0.00% | 0.00% | 0.00% | 0.00% | 0.00% |
| ICLR 2019 | ALLREJECT | 0 | 0 | 0 | 0 | 0 | 0 | 0 | 0 |
| ($n=1419$) | FORWARDREJECT | 0 | 0 | 0 | 0 | 0 | 0 | 0 | 0 |
| | **Ours** | 0 | 0 | 0 | 0 | 0 | 0 | 0 | 0 |
| | Relative Improvement (%) | 0.00% | 0.00% | 0.00% | 0.00% | 0.00% | 0.00% | 0.00% | 0.00% |
| ICLR 2020 | ALLREJECT | 1 | 0 | 0 | 0 | 0 | 0 | 0 | 0 |
| ($n=2213$) | FORWARDREJECT | 1 | 0 | 0 | 0 | 0 | 0 | 0 | 0 |
| | **Ours** | 1 | 0 | 0 | 0 | 0 | 0 | 0 | 0 |
| | Relative Improvement (%) | 0.00% | 0.00% | 0.00% | 0.00% | 0.00% | 0.00% | 0.00% | 0.00% |
| ICLR 2021 | ALLREJECT | 5 | 4 | 3 | 2 | 1 | 0 | 0 | 0 |
| ($n=2594$) | FORWARDREJECT | 5 | 4 | 3 | 2 | 1 | 0 | 0 | 0 |
| | **Ours** | 5 | 4 | 3 | 2 | 1 | 0 | 0 | 0 |
| | Relative Improvement (%) | 0.00% | 0.00% | 0.00% | 0.00% | 0.00% | 0.00% | 0.00% | 0.00% |
| ICLR 2022 | ALLREJECT | 0 | 0 | 0 | 0 | 0 | 0 | 0 | 0 |
| ($n=2617$) | FORWARDREJECT | 0 | 0 | 0 | 0 | 0 | 0 | 0 | 0 |
| | **Ours** | 0 | 0 | 0 | 0 | 0 | 0 | 0 | 0 |
| | Relative Improvement (%) | 0.00% | 0.00% | 0.00% | 0.00% | 0.00% | 0.00% | 0.00% | 0.00% |
| ICLR 2023 | ALLREJECT | 0 | 0 | 0 | 0 | 0 | 0 | 0 | 0 |
| ($n=3793$) | FORWARDREJECT | 0 | 0 | 0 | 0 | 0 | 0 | 0 | 0 |
| | **Ours** | 0 | 0 | 0 | 0 | 0 | 0 | 0 | 0 |
| | Relative Improvement (%) | 0.00% | 0.00% | 0.00% | 0.00% | 0.00% | 0.00% | 0.00% | 0.00% |
| ICLR 2024 | ALLREJECT | 16 | 11 | 8 | 7 | 6 | 5 | 4 | 3 |
| ($n=7404$) | FORWARDREJECT | 13 | 10 | 8 | 7 | 6 | 5 | 4 | 3 |
| | **Ours** | 12 | 10 | 8 | 7 | 6 | 5 | 4 | 3 |
| | Relative Improvement (%) | 7.69% | 0.00% | 0.00% | 0.00% | 0.00% | 0.00% | 0.00% | 0.00% |
| ICLR 2025 | ALLREJECT | 51 | 42 | 34 | 28 | 23 | 21 | 19 | 17 |
| ($n=11672$) | FORWARDREJECT | 47 | 39 | 31 | 26 | 23 | 21 | 19 | 17 |
| | **Ours** | 43 | 36 | 30 | 26 | 23 | 21 | 19 | 17 |
| | Relative Improvement (%) | 8.51% | 7.69% | 3.23% | 0.00% | 0.00% | 0.00% | 0.00% | 0.00% |

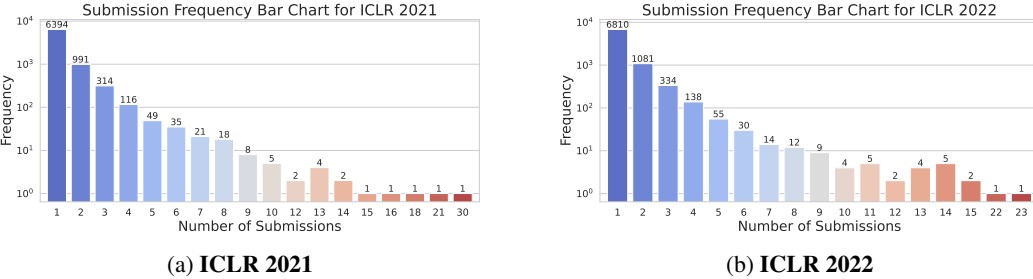

(a) **ICLR 2021**             (b) **ICLR 2022**

Figure 6: **Submission Frequency Bar Charts for ICLR 2021–2022.**

## LLM USAGE DISCLOSURE

LLMs were used only to polish language, such as grammar and wording. These models did not contribute to idea creation or writing, and the authors take full responsibility for this paper's content.

