# OpenReview forum: "Accept More, Reject Less: Reducing up to 19% Unnecessary Desk-Rejections over 11 Years of ICLR Data"
_ICLR.cc/2026/Conference — Submitted to ICLR 2026_

### Official Review · Reviewer_qWQd · 2025-10-15

**Soundness:** 4
**Presentation:** 4
**Contribution:** 2
**Rating:** 4
**Confidence:** 4

**Summary:**

Modern AI conferences often impose submission limits per author to manage the overwhelming number of paper submissions, which results in the desk-rejection of many papers. This paper seeks to minimize unnecessary rejections while adhering to these limits. Specifically, the authors formalize the desk-rejection process as an optimization problem where the variable is whether to desk-reject a paper, and the constraints ensure that no author exceeds their submission limit. They prove that this problem is computationally hard but propose an efficient algorithm based on linear programming relaxation and randomized rounding. They test their method on 11 years of ICLR submission data, demonstrating that it can preserve a non-negligible number of additional papers compared to existing policies, while being computationally efficient.

**Strengths:**

- The writing of this work is of high quality and its presentation is clear and well-structured. I found it easy to follow the problem formulation, the proposed solution, and the empirical evaluation.
- The evaluation on 11 years of real-world ICLR data is comprehensive and demonstrates that the proposed method can indeed reduce desk rejections while adhering to submission limits. The use of real conference data is a strong point.
- The authors' effort to improve conference peer review processes is commendable.

**Weaknesses:**

Despite the strengths, there are two major weaknesses that limit the impact of this work. I would take a more positive stance if any one of these issues were refuted, although I am currently confident in my assessment.
- The technical novelty of the proposed algorithm is limited. Without reading the actual method section, I can already tell from the introduction how the algorithm works. The use of linear programming relaxation and randomized rounding is a standard approach in combinatorial optimization, and the paper does not introduce any new techniques or insights in this regard.
- From a practical standpoint, optimizing the desk-rejection process may not be a priority for conference organizers. The submission limit policies are primarily designed to manage reviewer workload, and authors are expected to adhere to these limits at the time of submission. To a large extent, the responsibility lies with authors to manage their submissions. Therefore, while the proposed method can reduce unnecessary rejections, it may not significantly impact the overall conference review process or author experience.

As I said, any one of these issues alone does not warrant a negative assessment. However, together they lead me to believe that this work, while well-executed, does not make a substantial contribution to the field.

**Questions:**

- Are the above two weaknesses valid criticism of this work?

### Minor comments or questions that do not affect the overall evaluation

- Figure 2's font size is too small and should be increased for better readability.
- Apart from the fact that ICLR conference data is used, do you think this work fits well within the scope of ICLR?

---

### Official Review · Reviewer_9FkL · 2025-10-23

**Soundness:** 3
**Presentation:** 3
**Contribution:** 2
**Rating:** 4
**Confidence:** 3

**Summary:**

The number of submissions to peer-reviewed conferences, such as ICLR, has increased exponentially in the past few years. Conferences have started imposing a limit on the maximum submissions per author. Authors violating this constraint are asked to bring their submissions within the prescribed limit; otherwise, their papers beyond the limit get desk-rejected as a pre-processing step before the papers are submitted to paper-reviewer matching algorithms. Such desk rejection is typically posed as a constraint satisfaction problem (CSP). This paper argues that solving the CSP provides only a feasible solution, potentially compromising the fairness from the perspective of first-time authors whose papers may get desk rejected because of their senior co-authors violating the constraint. As an alternative, the paper proposes to minimise the total number of desk-rejected papers, converting the CSP to a constrained optimisation problem, posed as an ILP. To solve the ILP, they first solve its LP relaxation and propose a rounding algorithm that starts with the largest fractional assignment and converts it to 1 at the cost of other smaller fractional assignments of all the authors for whom the constraint is active.
The controlled experiments using past ICLR data show that the proposed method achieves up to 19.23% reduction (26 to  21) in desk rejects for ICLR 2024 when max author limit = 22.

**Strengths:**

1. The paper studies a pertinent issue - authors' welfare (especially junior researchers’) in the current peer-review system.
2. The paper proposes a natural extension of the current CSP formulation by adding an objective function to it that can potentially improve author welfare.
3.  Experimental setup is extensive -- the paper experiments with 11 years of ICLR data and different thresholds for max. submissions per author.

**Weaknesses:**

1. Lack of baselines:

	a. There are other works ( cited in the paper ) that address the same issue, e.g. Cao et al (2025) also propose an objective function to address fairness in desk rejects due to max submission limits. However, it has not been used as a baseline.

	b. There exist many heuristics to solve  ILPs, eg, Branch and Bound also involve solving the LP relaxation and then tackling the fractional assignments. Wouldn’t the existing black-box ILP solvers be able to solve the proposed ILPs? There is no comparison to those. They may be costly, but this desk rejection ILP is not the bottleneck in the overall paper-reviewer assignment process.


2. Issues with the metric: if the baseline method desk-rejects 2 papers and your method rejects only 1, then according to the used metric, it would be 50% improvement! Please consider reporting the confidence intervals.


3. Subjective analysis of the additional papers that are not rejected by the proposed algorithm is missing -- some intuitive discussion on why and how the additional accepted papers promote author’s welfare would be very useful.

Overall, there are two main contributions of the paper: (a) adding an objective function to CSP, and (b) the rounding algorithm to convert an LP solution to an integral solution. In light of the above weaknesses, I am not sure if they are novel enough.

**Other minor comments:**

Line 105-106:  The paper argues that desk-rejection mechanisms are proposed to reduce the reviewer workload in peer review. I think this is not the only objective of desk rejects.  If this were correct, then your proposed algorithm is actually doing the opposite -- its rejecting less, thereby increasing the load on reviewers. I think the primary objective of desk-rejects is fairness to the system and rules.

Line 144-146: Definition 3.1 introduces $x$ as a desk-rejection vector, whereas it is actually a vector of acceptance. Please consider clarifying it early to improve the readability.

**Questions:**

Line 14 in Algo 3: I think there should be a condition on the cardinality of $S_i$ as well. Otherwise, wouldn’t $S_i  = S \cap T_i$ also satisfy the condition $\sum_{j \in S_i} \tilde{x}_j \ge (1- x_l)$ ?   Also, is it guaranteed that such an $S_i$ would always exist?
In general, I think we need a little more discussion and intuitive analysis of the proposed Algo 3.

---

### Official Review · Reviewer_U3uH · 2025-10-28

**Soundness:** 2
**Presentation:** 3
**Contribution:** 1
**Rating:** 2
**Confidence:** 4

**Summary:**

This paper addresses the problem of desk rejections at AI conferences that implement per-author submission caps. It argues that common heuristic-based rejection policies (e.g., based on submission ID) are suboptimal. The authors formulate the task of selecting which papers to reject as an integer linear programming problem with the objective of maximizing the total number of papers that are desk-accepted for review. To solve this, they propose a two-step method involving an LP relaxation followed by a custom rounding algorithm to ensure all constraints are met. Using 11 years of ICLR submission data for simulation, the authors report that their method can reduce the number of desk-rejected papers by up to 19.23% compared to baseline policies, while running in a computationally efficient manner.

**Strengths:**

* **Addresses a relevant problem area**: The paper correctly identifies that desk-rejection policies at top conferences are a significant issue deserving of systematic study and improvement.
* **Achieves its core objective**: The paper successfully develops a solution that directly addresses the problem it formulates.
* **Empirical evidence**: The empirical validation on a large-scale, real-world dataset from 11 years of ICLR submissions is a strong point and allows for a convincing demonstration of the algorithm's behavior.

**Weaknesses:**

* **Limited objective function**: The paper's primary weakness, which undermines its entire contribution, is its choice of objective function. The authors equate "improving author welfare" with a simplistic, utilitarian goal: maximizing the raw number of desk-accepted papers. This premise is fundamentally flawed, as it ignores the more critical goals of a fair academic review process, such as promoting author diversity, protecting the work of early-career researchers, and ensuring a variety of institutions are represented. A policy optimized for paper throughput could easily and systematically disadvantage researchers from smaller labs in favor of large, prolific groups, thereby harming the very community it claims to help.
* **Questionable practical impact**: The practical significance of the proposed solution is questionable due to the nature of existing submission policies. Many top-tier conferences that enforce limits set a high bar (e.g., 25 for CVPR 2025), and since these limits are well-publicized, authors are strongly incentivized to self-regulate and avoid violations. This suggests the algorithm may only apply to a small number of edge cases. The paper's own results support this concern: in the simulation for ICLR 2025 with a limit of b=25, the proposed method saves only 4 papers over the ForwardReject baseline. This marginal improvement for realistic, high-limit scenarios raises doubts about the algorithm's real-world impact.

**Questions:**

* **Justification of the objective**: Can you provide a strong justification for why maximizing the raw count of papers is a more desirable goal for the AI community than maximizing a metric of diversity, such as the number of unique authors or unique affiliations represented in the post-rejection pool?
* **Potential for negative impact**: Your work claims to improve "author welfare". However, your objective function could systematically favor large, highly-prolific research groups over smaller labs or early-career researchers. Have you analyzed whether your method disproportionately benefits certain author demographics? How do you reconcile the potential for your method to reduce diversity with your claim of improving the system?
* **ICLR as a case study**: Do you have an estimate for the value of b for ICLR?

---

### Official Review · Reviewer_ovRh · 2025-11-02

**Soundness:** 3
**Presentation:** 3
**Contribution:** 2
**Rating:** 4
**Confidence:** 4

**Summary:**

Many AI venues impose a per-author submission cap and desk-reject excess papers, often by ID order. The paper asks: given an authorship matrix and a per-author cap b, can we minimize unnecessary desk rejections while respecting caps? The authors formalize this as selecting a subset of papers that maximizes the number sent to review subject to per-author capacity constraints. They propose a two-stage algorithm: (1) solve the LP relaxation max 1^T x subject to Ax <= b 1, x in [0,1]^m; (2) apply a deterministic rounding routine that iteratively rounds the largest fractional variable to 1 and zeros out certain other fractional variables to restore feasibility for each co-author. Using OpenReview data for ICLR 2013-25 they simulate submission-limit desk rejection for a range of b and compare their method to two baselines that emulate common policies. They report consistent reductions in desk rejections.

**Strengths:**

(+) This is a timely and well-scoped problem with practical impact. The paper targets a policy that now affects many large CS conferences. Formalizing the desk-rejection step as a clean packing LP/ILP is useful and portable. Furthermore, the proposed method amounts to a simple algorithm that a program chair could conceivably implement.
(+) Across 10+ years of ICLR submissions, the method reduces desk rejections vis-a-vis the operational baselines.

**Weaknesses:**

There is a potential technical issue with the MAXROUNDING algorithm. If my reading is correct: after rounding a paper $l$ up to 1, the algorithm checks each co-author $i$ and, if  $ \sum_{j\in T_i} \tilde{x}_j > b $ , it finds a set $S_i \subset (S \cap T_i)$ whose fractional mass totals at least $ (1-x_l) $ and zeros them out.

This choice of required mass $(1-x_l)$ is not minimal and be strictly larger than the actual overflow
$\delta_i = \max [0, \sum_{j \in T_i} \tilde{x}_j - b ] $.

There are feasible states where $\delta_i < (1-x_l)$ yet the residual fractional mass $\sum_{j \in S \cap T_i} \tilde{x}_j$ is less than $(1-x_l)$ (e.g., when author $i$ is already near $b-1$ with a small remaining fractional slack).

In such a case Algorithm 3 (as written), may fail to find $S_i$ despite the fact that only $\delta_i$ needs to be deleted to restore feasibility. The proof of Theorem 4.6 does not argue existence of $S_i$ under the stronger $(1-x_l)$ requirement; it simply asserts that "a sufficient number of papers" are rejected, which does not address the mismatch between the needed mass $\delta_i$, and the requested mass $(1-x_l)$.

I believe this is an important gap, but probably fixable if one was to replace line 14 of the Algorithm with $\sum_{j \in S_i} \geq \delta_i$ and add a short invariant-based argument in the proof in the appendix.

**Questions:**

Please address the above described issue.

---

### Official Review · Reviewer_wuRA · 2025-11-04

**Soundness:** 3
**Presentation:** 3
**Contribution:** 3
**Rating:** 6
**Confidence:** 4

**Summary:**

The paper considers the problem of whether it is possible to follow submission limits while minimizing needless rejections. The paper formalizes it as a discrete optimization problem and proposes a new algorithm designed to satisfy the per-author limits while maximizing the total number of papers that can be "desk-accepted" for review. Testing their method on 11 years of ICLR submission data, they demonstrate it can preserve about 20% more papers than current policies.

**Strengths:**

The paper proposes a principled solution to understand whether it is possible to follow submission limits while minimizing needless rejections. To achieve efficient computation, the paper propose a two-stage solution that firstly solves the linear program relaxation of the original integer program, and then converts the fractional solution to a provably feasible integer solution via a specific rounding scheme.

**Weaknesses:**

While the paper formalizes the problem, it arguably oversimplifies it. A more realistic model would allow for variable submission limits based on factors like an author's seniority or research area.

**Questions:**

Can the solution be easily generalized to the setting where there are different groups of authors who are given different submission limits?

---

### Meta-Review · Area_Chair_rJwc · 2026-01-06

**Summary:**

1.  Technical Flaws in MAXROUNDING algorithm
2. Fundamentally Flawed Objective Function
3. Limited Technical Novelty
4.  Lack of Baselines

**Reviewer Concerns:**

No rebuttal submitted. Most of the concerns are still valid.

**Reviewer Scores:**

Likely to maintain the scores.

---

### Decision · Program_Chairs · 2026-01-26

Reject